# Irradiation by a Combination of Different Peak-Wavelength Ultraviolet-Light Emitting Diodes Enhances the Inactivation of Influenza A Viruses

**DOI:** 10.3390/microorganisms8071014

**Published:** 2020-07-08

**Authors:** Mizuki Kojima, Kazuaki Mawatari, Takahiro Emoto, Risa Nishisaka-Nonaka, Thi Kim Ngan Bui, Takaaki Shimohata, Takashi Uebanso, Masatake Akutagawa, Yohsuke Kinouchi, Takahiro Wada, Masayuki Okamoto, Hiroshi Ito, Kenji Tojo, Tomo Daidoji, Takaaki Nakaya, Akira Takahashi

**Affiliations:** 1Department of Preventive Environment and Nutrition, Institute of Biomedical Sciences, Tokushima University Graduate School, Kuramoto-cho 3-18-15, Tokushima City, Tokushima 770-8503, Japan; mizugojira@gmail.com (M.K.); st_2r_flight062b@yahoo.co.jp (R.N.-N.); kimnganvdd1190@gmail.com (T.K.N.B.); shimohata@tokushima-u.ac.jp (T.S.); uebanso@tokushima-u.ac.jp (T.U.); akiratak@tokushima-u.ac.jp (A.T.); 2Graduate School of Technology, Industrial and Social Sciences, Minamijyousanjima-cho 2-1, Tokushima City, Tokushima 770-8506, Japan; emoto@tokushima-u.ac.jp (T.E.); makutaga@tokushima-u.ac.jp (M.A.); kinouchi@tokushima-u.ac.jp (Y.K.); 3Nihon Funen Co., Ltd., 179-1 Mitsujima-shinden, Kawashima-cho, Yoshinogawa City, Tokushima 779-3394, Japan.; t-wada@nihonfunen.co.jp (T.W.); okamoto@nihonfunen.co.jp (M.O.); ito@nihonfunen.co.jp (H.I.); tojyo@nihonfunen.co.jp (K.T.); 4Department of Infectious Diseases, Graduate School of Medical Science, Kyoto Prefectural University of Medicine, 465 Kajii-cho, Kawaramachi-Hirokoji, Kamigyo-ku, Kyoto 602-8566, Japan; daidoji@koto.kpu-m.ac.jp (T.D.); tnakaya@koto.kpu-m.ac.jp (T.N.)

**Keywords:** light emitting diode, ultraviolet, influenza A virus

## Abstract

Influenza A viruses (IAVs) pose a serious global threat to humans and their livestock. This study aimed to determine the ideal irradiation by ultraviolet-light emitting diodes (UV-LEDs) for IAV disinfection. We irradiated the IAV H1N1 subtype with 4.8 mJ/cm^2^ UV using eight UV-LEDs [peak wavelengths (WL) = 365, 310, 300, 290, 280, 270, and 260 nm)] or a mercury low pressure (LP)-UV lamp (Peak WL = 254 nm). Inactivation was evaluated by the infection ratio of Madin–Darby canine kidney (MDCK) cells or chicken embryonated eggs. Irradiation by the 260 nm UV-LED showed the highest inactivation among all treatments. Because the irradiation-induced inactivation effects strongly correlated with damage to viral RNA, we calculated the correlation coefficient (*R*_AE_) between the irradiant spectrum and absorption of viral RNA. The *R*_AE_ scores strongly correlated with the inactivation by the UV-LEDs and LP-UV lamp. To increase the *R*_AE_ score, we combined three different peak WL UV-LEDs (hybrid UV-LED). The hybrid UV-LED (*R*_AE_ = 86.3) significantly inactivated both H1N1 and H6N2 subtypes to a greater extent than 260 nm (*R*_AE_ = 68.6) or 270 nm (*R*_AE_ = 42.2) UV-LEDs. The *R*_AE_ score is an important factor for increasing the virucidal effects of UV-LED irradiation.

## 1. Introduction

Influenza viruses are enveloped viruses whose genome consists of segmented negative-sense single-strand RNA segments. There are four types of influenza viruses, including A, B, C and D, and types A and B show epidemic spread. Influenza A viruses (IAVs) are encircled by the M1 matrix protein and a host-derived lipid bilayer envelope in which the virus surface glycoproteins hemagglutinin (HA) and neuraminidase (NA), as well as the M2 matrix protein, are embedded [1]. HA and NA play critical roles in viral entry into host cells and release from the cells, respectively [2]. IAVs bind to sialic acid molecules, such as α2,6-linked sialo-glycans (SAα2,6), which are abundant in the human upper respiratory tract [3]. Following internalization, receptor-bound viruses are delivered to endosomes, and upon acidification, HAs are activated to fuse the virus and endosomal membranes [4]. NA cleaves the sialic acid molecule, thereby freeing the virus and allowing it to infect other cells in the host organism.

IAVs exhibit zoonotic potential because novel IAV mutants frequently emerge and are able to cross species barriers. Therefore, IAV infections are a globally important issue not only in humans but also in their livestock. Avian influenza (AI) viruses are classified by their pathogenic strength based on the feasibility of infection in chickens [5]. The critical genetic difference determining the low pathogenic (LPAI) or highly pathogenic (HPAI) phenotype depends on the HA cleavage site [6]. HPAI viruses, such as H5N1 and H7N9, have only two HA subtypes, including H5 and H7 [6]. In addition to infecting chickens, H5N1 subtypes emerged as a human pathogen in 1997 with the expected potential to undergo sustained human-to-human transmission and pandemic viral spread [7]. LPAI H6 subtypes have a broader host range than any other IAV subtype [8] and are suggested to be involved in the generation of human H5N1, H9N2, and H5N6 subtypes [7,9]. H6 subtype reassortment viruses might have crossed the species barrier and infected mammals, including humans, without adaptation. Recent seroprevalence research showed seropositivity for H6 viruses among occupational exposure workers in 19 provinces of China [10], and the analysis of veterinarians exposed to birds showed that the H6-specific antibody was significantly elevated in the United States [11]. In 2013, a H6N1 virus was isolated in Taiwan from a 20-year-old woman with symptoms, including fever, cough, headache, and muscle aches [12]. The study of H6 viruses isolated from patients in Taiwan in the past 14 years suggests an elevated threat of H6 viruses to human health [13]. These data indicate that in addition to HPAI H5 and H7 subtypes, H6 subtype IAVs might pose a potential threat to human health.

To prevent viral infections, alcohol is generally used to inactivate influenza viruses but has the disadvantage that it cannot be used for livestock or food. Therefore, chlorination, ozonation, and UV light are widely used for disinfection. However, there are some health concerns regarding the use of chemicals-based disinfection methods. For instance, residual chlorine in drinking water can cause the formation of potentially carcinogenic halogenated by-products [9]. In addition, ozonation can lead to excess concentrations of undesired by-products, such as bromate, which is a potential human carcinogen [14,15]. In contrast, sunlight or ultraviolet (UV) irradiation does not produce residual chemicals. UV rays can be classified by wavelength into UVA (320–400 nm), UVB (280–320 nm), and UVC (< 280 nm). A low-pressure UV (LP-UV) mercury lamp radiates a monochromatic peak wavelength (254 nm) and is used for common water treatment processes to remove and inactivate viral and microbial pathogens, mainly by damaging their genome [12]. In recent years, UV-light emitting diodes (UV-LEDs) have been developed as an alternative UV light emission source. While mercury lamps only emit light at one wavelength or over a broad range of wavelengths, UV-LEDs can irradiate light of a single wavelength without a filter and are capable of emitting light at multiple individual wavelengths [13]. LEDs are created by connecting p- and n-type semiconductors that move electrons into positively charged holes between these two materials. The wavelength of light will depend on the type of material used for the two semiconductors. Indium gallium nitride (InGaN) is a widely available semiconductor for highly efficient blue light or wavelengths in the UVA range. InGaN-based UVA-LEDs exhibit almost 50% of external quantum efficiencies but cannot radiate deep UV wavelengths. Recently, aluminum gallium nitride (AlGalN) has been used as a semiconductor for deeper UV wavelengths from 250 to 350 nm and manufactured for many potential applications, including microbial disinfection. Kim D.K., et al. [16] demonstrated that irradiation by 266, 270, 275, and 279 nm UV-LEDs inactivated both gram-positive and gram-negative bacteria and yeast. They showed that a low WL UV-LED induced higher microbial reduction of both bacteria and yeasts than a high WL one, but there was no significant difference (*p* > 0.05). Meanwhile, Li Q.G., et al. [17] reported that a 265 nm UV-LED was more effective than a 280 nm UV-LED or LP-UV lamp for *E**scherichia coli* inactivation. For the inactivation of bacteriophages, Kim D.K., et al. [18] reported that a 266 nm UVC-LED was more effective than a 279 nm UV-LED or LP-UV lamp. Sholtes K.A., et al. [19] reported that the microbial inactivation kinetics of the 260 nm UV-LED were not significantly different than those of the LP-UV lamp for *E. coli* and bacteriophages but were higher than LP-UV lamps for *Bacillus atrophaeus* spores. From these reports, low WL UV-LEDs around 260–266 nm would exhibit the highest microbial reduction among UV-LEDs and LP-UV lamps. We previously reported that irradiation by UVA (365 nm), UVB (310 nm), and UVC (280 nm)-LEDs inhibited the infectious titer of both IAV H1N1 and H5N1 subtypes [20]. In our study, we found that a UVC-LED with a peak WL of 280 nm showed the highest inactivation effect on IAVs among the UV-LEDs tested. However, we did not compare the inactivation effect of IAVs between UV-LEDs and LP-UV lamps, and how the peak WL of UV-LED is most effective for the inactivation of IAVs remains unclear. To investigate the highest inactivation effect of IAVs in this study, we irradiated virus suspensions of the H1N1 subtype using eight UV-LEDs with peak wavelengths of 365, 310, 300, 290, 280, 270, and 260 nm and measured the infection ratio using different host organisms, including Madin–Darby canine kidney (MDCK) cells and chicken embryonated eggs. In addition, we compared the inactivation effect of UV-LEDs with that of an LP-UV lamp. We found that the emission spectrum of UV-LEDs, which showed the highest inactivation effect among all treatments, strongly correlated with the absorbance spectrum of viral RNA. Therefore, we demonstrated that irradiation with the spectrum that strongly correlates with the absorption spectrum of viral RNA by the combination of different peak-wavelength UV-LEDs (hybrid UV-LED) enhanced the inactivation of both H1N1 and H6N2 subtypes.

## 2. Materials and Methods 

### 2.1. Cells and Virus Strains

MDCK cells and the IAV H1N1 subtype (strain A/Puerto Rico/8/1934) were a kind gift from Professor Akio Adachi (Tokushima University Graduate School). The LPAI H6N2 subtype (A/Duck/Hong Kong/960/1980) was obtained from the Research Foundation for Microbial Diseases at Osaka University. MDCK cells were maintained at 37 °C in a humidified atmosphere containing 5% CO_2_ and cultured in Dulbecco’s modified Eagle’s medium (DMEM) supplemented with 5% fetal bovine serum (AusGeneX, Oxenford, Australia) and 60 µg/mL kanamycin (Fujifilm Wako Pure Chemical Industries, Ltd., Tokyo, Japan). The virus suspension was prepared for irradiation experiments by propagating the IAV subtypes in 10-day-old chicken embryonated eggs (Ishii Poultry Agricultural Cooperative, Tokushima, Japan) for 48 h at 37 °C.

### 2.2. UV-LEDs and LP-UV Lamp Irradiation of the Virus Suspensions

Eight different peak wavelength UV-LEDs (Nichia, Tokushima, Japan) and an LP-UV lamp were used to irradiate the viral suspensions in this study (Table 1 and Figure 1). The three individual LEDs were on a printed circuit board (Audio-Q, Shizuoka, Japan, Figure 1a) and connected in series to a current-controlling single power source (PAS40-9, Kikusui Electronics Corp., Kanagawa, Japan). All UV-LEDs controlled the forward current (IF) by the power source for the adjustment of the fluence rate (2.4 mW/cm^2^). A volume of 0.3 mL virus suspension with an infectivity titer of 1.92 ± 0.13 × 10^7^ (H1N1) or 0.83 ± 0.05 × 10^7^ (H6N2) focus-forming units (FFU)/mL was placed in a stainless steel cylinder cup (10 mm diameter and depth). The UV-LEDs and LP-UV lamp were emitted downward onto the surface of the solution for 2 sec (fluence = 4.8 mJ/cm^2^). The spectral fluence rates on the surface of samples were measured using an MCPD 3700A multiple wavelength photometer (Otsuka Electronics, Osaka, Japan) and PMA-12 Photonic multichannel analyzer (Hamamatsu photonics, Shizuoka, Japan).

### 2.3. Infection of MDCK Cells and the Focus-Forming Assay

MDCK cells were cultured in 48-well plates. At 13 h post-infection with UV-irradiated virus suspensions, the cells were fixed for 30 min at room temperature with buffered 4% paraformaldehyde and washed three times with phosphate-buffered saline (PBS). An unirradiated virus suspension was used as a control. Then, the cells were stained with a rabbit polyclonal antibody against the LPAI virus H5N2 subtype (strain A/duck Hong Kong/342/1978) to detect influenza virus antigens. This antibody recognizes the influenza virus NP and M1 proteins [21]. Antibody-binding viral proteins were detected with an Alexa Fluor 488-conjugated secondary antibody (Molecular Probes, Carlsbad, CA, USA) diluted 1:500 in PBS containing 1% bovine serum albumin. An IX71N fluorescent microscope (Olympus, Tokyo, Japan) was used to observe the cells and count the FFUs. The effects of the LED irradiations on viral inactivation were determined by the infection ratio (log_10_ FFU ratio), which was calculated as log_10_ FFU ratio = log_10_ (*Nt*/*N_0_*), where *N_t_* is the FFU of the UV-irradiated sample, and *N_0_* is the FFU of the sample without UV irradiation.

### 2.4. Infection of Embryonated Chicken Eggs

After the irradiation of virus suspensions with the UV-LEDs and LP-UV lamp, the suspensions were diluted at 1:10^1^–1:10^8^ and infected into 10-day-old chicken embryonated eggs. Following incubation at 36 °C for 72 h, the allantoic fluids of the eggs were collected, clarified by centrifugation, and analyzed by HA assays. Viral titers were calculated as the 50% embryo infection dose (EID_50_) as described by Guan J., et al. [22], and the effects of the LED irradiations were determined by the infection ratio (log_10_ EID_50_ ratio) calculated as log_10_ EID_50_ ratio = log_10_ (*N_t_*/*N_0_*), where *N_t_* is the EID_50_ of the UV-irradiated sample, and *N_0_* is the EID_50_ of the sample without UV irradiation.

### 2.5. HA Assay

The HA titer of the chorioallantoic fluid from the embryonated chicken eggs and the virus suspensions with or without UV irradiation were measured using a standard HA assay [23]. Two-fold serial dilutions of virus suspensions with or without UV irradiation were added to a round-bottomed 96-well plate and then mixed with 0.5% chicken red blood cells (Kohjin Bio Co., Ltd., Saitama, Japan). After incubation for 3 h at room temperature, HA titers were determined as the highest dilution at which complete agglutination was observed.

### 2.6. RNA Extraction and RT-qPCR

The damage to viral RNA by UV irradiation was measured by reverse transcription (RT)-quantitative real-time polymerase chain reaction (qPCR) as previously reported [20,24,25]. To isolate viral RNA from the viral suspension with or without UV irradiation, a QIAamp Viral RNA Mini Kit (Qiagen, Redwood City, CA, USA) was used according to the manufacturer’s instructions. Specific RT of vRNA of H1N1 segment 6 was performed with tagged-primers (5’-GGCCGTCATGGTGGCGAATACTATAATGACTGATGGCCCGAGT-3’) using Superscript III reverse transcriptase (Invitrogen, Carlsbad, CA, USA), as previously reported by Kawakami E., et al. [26]. A 5.5 µL mixture containing viral RNA and 10 pmol of the tagged primer was heated for 10 min at 65 °C and then chilled immediately on ice for 5 min. Next, 14.5 µL of a preheated reaction mixture [4 µL First Strand buffer (Invitrogen), 1 µL 0.1 M dithiothreitol (Invitrogen), 4 µL dNTP mix (10 mM each, TaKaRa Bio, Shiga, Japan), 1 µL Superscript III reverse transcriptase (200 U/µL, Invitrogen), 1 µL RNasin Plus RNase inhibitor (40 U/µL, Promega, Fitchburg, WI, USA), and 3.5 µL DEPC water] were added and incubated at 60 °C for 1 h. Real-time PCR was performed with a SYBER Green Premix Ex Taq II (Takara Bio) using a real-time PCR system Light cycler^®^ 2.0 Instrument (Roche, Mannheim, Germany). cDNA (1.2 µL) was added to the qPCR reaction mixture [6 µL SYBER Green Premix Ex Taq II, 0.24 µL forward primer (5’-GGCCGTCATGGTGGCGAAT-3’, 10 µM), 0.24 µL reverse primer (5’-ACATCACTTTGCCGGTATCAGGGT-3’, 10 µM), and 4.32 µL double-distilled water]. The reactions were heated at 95 °C for 30 seconds followed by 40 cycles of a 10-second denaturing step at 95 °C, 20-second annealing step at 60 °C, and 15-second extension step at 72 °C.

### 2.7. Calculation of Correlation Coefficients between the Absorbance Spectrum of Viral RNA and Emission Spectrum of UV-LEDs

We were interested in investigating the relationship between the absorbance spectrum of viral RNA and the UV-LED emission spectra. After propagating the virus in 10-day-old chicken embryonated eggs for 48 h at 37 °C, allantoic fluids were precleared by centrifugation at 3300× *g* for 20 min followed by filtration through 0.45 ìmö pore filters. The viruses were then purified by centrifugation (112,500× *g* for 2 h) through PBS containing 20% sucrose, as previously reported [21]. Virus pellets were resuspended in PBS and purified the viral RNA by a QIAamp Viral RNA Mini Kit (Qiagen). The purified viral RNA were checked by RT-PCR for segment 4 (Appendix A), as previously reported [20]. Although the absorbance spectrum of IAV RNA was measured with a UV-VIS spectrophotometer DU730 (Beckman Coulter, Brea, CA, USA), there is a difference in the spectral resolution between the spectrophotometer and the photonic multichannel analyzer (Hamamatsu photonics, Shizuoka, Japan) used for the measurement of emission spectra. For computation, it is often necessary to convert them to the same spectral resolution in the same range of wavelengths (245 nm to 320 nm). Therefore, in this work, the wavelength resolution of the spectrum was re-sampled into the same resolution (0.1 nm) using cubic spline interpolation. Further, the spectra were normalized by the maximum value in each spectrum. We used the measurement *R*_AE_ to investigate the relationship between the normalized absorbance spectrum of viral RNA and the normalized emission spectrum. *R*_AE_ was calculated using the following formula:(1)$$R_{\mathrm{AE}}=\sum_{\lambda =b}^{B}{\left|A\left(\lambda \right)\cdot S\left(\lambda \right)\right|}^{2}$$
where *A*(λ) is the normalized viral RNA absorption spectrum, and *S*(λ) is the normalized emission spectrum of the UV-LED or LP-UV lamp. The wavelength of the spectrum was used from 245 nm (b) to 320 nm (B) for the calculation. For the calculation of *R*_AE_ of the 365 nm UV-LED, the maximum wavelength ranged up to 400 nm and was normalized by those of the other UV-LEDs. We used MATLAB R2017b software (The Mathworks, Natick, MA, USA) for the computation of *R*_AE_.

### 2.8. Statistical Analysis

Statistical analysis of differences was performed using ANOVA with Bonferroni’s multiple comparison tests using Statview 5.0 software (SAS Institute Inc., Cary, NC, USA). The Student’s t-test was used for paired data where appropriate. *p* < 0.05 was considered statistically significant. Spearman’s rank correlation test was used for the analysis of the association between two variables. For comparisons between two correlations, the correlation coefficient values were transformed into z scores by Fisher’s r to z transformation and analyzed for statistical significance by determining the observed z test statistic.

## 3. Results

### 3.1. Inactivation Effects of Irradiations by Different UV-LEDs and a Low-Pressure UV Lamp on the IAV H1N1 Subtype

To determine the effects of UV irradiations on the IAV H1N1 subtype, we irradiated virus suspensions using UV-LEDs or an LP-UV lamp with an equal fluence rate (2.4 mW/cm^2^) and duration (2 sec) and infected MDCK cells and embryonated chicken eggs with these virus suspensions. Irradiation by 290–365 nm UV-LEDs did not affect the infection ratio, whereas the 260–280 nm UV-LEDs and LP-UV lamp decreased the ratio in both MDCK cells and embryonated chicken eggs (Figure 2). These results suggested that low fluence irradiations (4.8 mJ/cm^2^) by 260–280 nm UV-LEDs showed an inactivation effect on the IAV H1N1 subtype in both host organisms. The lowering effect of UV irradiations on the infection ratio was correlated with those peak WLs, but the lowering effect on the infection ratio by the 260 nm UV-LED was significantly lower than that by the LP-UV lamp.

### 3.2. Effect of UV-LED Irradiation on the HA Activity of Viral Suspensions

HA plays an important role in the attachment of viruses to the cell surface of the host and release of the viral genome into the cytoplasm [27]. To determine the effect of UV-LEDs on the HA of the H1N1 subtype, we measured the HA activity of the virus suspension in the presence or absence of 4.8 mJ/cm^2^ UV irradiation. None of the UV-LEDs or the LP-UV lamp altered the HA activities (Figure 3), suggesting that the UV irradiations at 4.8 mJ/cm^2^ did not affect the binding capacity of IAV to host cells.

### 3.3. UV-LED- and LP-UV Lamp-Induced Damage to Viral RNA 

To measure the viral RNA damage induced by LED irradiation, we purified viral RNA from viral suspensions with or without UV-LEDs and LP-UV lamp irradiation at 4.8 mJ/cm^2^, and performed strand-specific RT-qPCR, as described in the Materials and Methods section [20,24,25]. No effect was observed with 300–365 UV-LEDs, whereas 260–290 nm UV-LED and LP-UV lamp irradiation decreased the relative level of viral RNA, suggesting that 260–290 nm UV-LEDs could damage viral RNA (Figure 4a). Because the 260 nm UV-LED had the highest effects on both the disinfection ratio and damage of viral RNA, we assessed the correlation between the effect on the disinfection ratio and damage to viral RNA by the UV irradiations (Figure 4b). The ratios of damage to viral RNA by the UV irradiations were strongly correlated with the disinfection ratio in MDCK cells (*R* = 0.9524) and embryonated chicken eggs (*R* = 0.9102). These results suggest that the inactivation effects of irradiation by the UV-LEDs and LP-UV lamp are dependent on the damage of viral RNA, and we hypothesized that the wider range of emission spectra of UV-LEDs than that of the LP-UV lamp was an important factor for the inactivation of IAVs and induction of RNA damage.

### 3.4. R_AE_ between the Absorbance Spectrum of Viral RNA and Emission Spectrum of UV Irradiations

Next, we determined the relationship between the absorbance spectrum of viral RNA and the emission spectrum of UV irradiations. To investigate the relationship, we purified viral RNA from the chorioallantoic fluid in IAV-infected embryonated chicken eggs and measured the absorbance spectrum (Figure 5). We then calculated the *R*_AE_ between the spectrum and emission spectrum of the UV-LEDs and LP-UV lamp (Figure 5b and Table 1). The *R*_AE_ scores of 300–365 nm UV-LEDs were lower than 1.5. In addition, the lower the peak WL of 260–290 nm UV-LEDs was, the higher the *R*_AE_ was. The *R*_AE_ score of the LP-UV lamp (*R*_AE_ = 11.15) was lower than that of the 260 nm UV-LED (*R*_AE_ = 68.56) and 270 nm UV-LED (*R*_AE_ = 42.24) (Figure 5b). The correlation coefficient between the disinfection ratio and *R*_AE_ of UV irradiations was significantly higher than that between the disinfection ratio and peak WL of the irradiations (Figure 6 and Table 2). From these results, *R*_AE_ is a more important index for the inactivation of IAVs by UV irradiation than WL.

### 3.5. Inactivation Effects of Hybrid UV-LED Irradiations on IAV H1N1 and H6N2 Subtypes

To demonstrate an inactivation effect of UV irradiation with a higher *R*_AE_, we combined three different UV-LEDs (hybrid LED), including MO-2257-U270 (Table 1), MO-2257-U260 (Table 1), and one whose peak WL was 258.5 nm, which was recently developed by Nichia (Tokushima, Japan). We simultaneously irradiated at 2.4 mW/cm^2^ (0.8 mW/cm^2^ each UV-LED) (Figure 7a,b). The *R*_AE_ score of the hybrid UV-LED (*R*_AE_ = 86.3) was higher than those of the other UV irradiations in this study (Figure 7b).

To compare the inactivation effect of the hybrid UV-LED and 258 nm, 260 nm, or 270 nm UV-LED on IAVs, we irradiated virus suspensions of both H1N1 and H6N2 subtypes and infected them into MDCK cells and embryonated chicken eggs. In infected MDCK cells, the hybrid LED had a significant (*p* < 0.05) or a moderate trend toward significantly (*p* = 0.068) higher inactivation effects on both H1N1 and H6N2 subtypes than the 258 nm, 260 nm or 270 nm UV-LED (Figure 7c). In infected embryonated chicken eggs, the hybrid LED showed a significant (*p* < 0.05) or a moderate trend toward significantly (*p* = 0.067) higher inactivation effect on H1N1 subtype than the 258 nm, 260 nm, or 270 nm UV-LED, but the effect on H6N2 was not different between the hybrid LED and 258 nm LED (Figure 7d).

## 4. Discussion

In this study, we investigated the peak WL of UV-LEDs with the highest viral inactivation effects on IAV and compared the inactivation effect between the UV-LEDs and LP-UV lamp. The 260 nm UV-LED exhibited the highest effect among the UV-LEDs and LP-UV lamp at 4.8 mW/cm^2^ for the inactivation of the H1N1 subtype infected into MDCK cells and chicken embryonated eggs. Our data are supported by two reports on the virucidal effects of irradiation on bacteriophages. Kim D.K., et al. [18] reported that the 266 nm UV-LED had a higher inactivation effect on non-envelope bacteriophages, including MS2, Qβ, and ΦX174, than the LP-UV lamp. In addition, Beck S.E., et al. [28] reported that the 260 nm UV-LED had a higher inactivation effect on MS2 bacteriophages than the LP-UV lamp. Furthermore, we demonstrated that the viral inactivation effects of UV-LEDs at 4.8 mW/cm^2^ on IAV were not due to changes in the envelope protein HA. These data suggest that the beneficial effect of UV-LEDs compared with the LP-UV lamp is not due to envelope damage. Irradiation by the LP-UV lamp is well known to inactivate viral and microbial pathogens by mainly damaging their genome [12]. By vRNA strand-specific RT-qPCR, we found that irradiation with the 260 nm UV-LED damaged viral RNA to a greater extent than the LP-UV lamp (Figure 4a), and the damage to viral RNA by UV irradiation was strongly correlated to the inactivation effect (Figure 4b). Genomic RNA from MS2 bacteriophages has an absorbance spectrum between 240 nm and 300 nm [24], which is similar to the IAV RNA that we measured in this study (Figure 5b). These data are supported by a previous study that showed RNA damage by UV irradiation closely mirrored the loss of infectivity in MS2 bacteriophages [24]. The LP-UV lamp emits monochromatic UV irradiation at a wavelength of 254 nm. In contrast, the 260 nm UV-LED irradiates at a wavelength near the LP-UV lamp but at a wider range of wavelengths (Figure 1c and Figure 5b). From these data, we assessed whether the viral inactivation effect on IAVs by UV irradiation was due to the *R*_AE_ between the absorbance spectrum of viral RNA and the emission spectrum of the UV-LEDs and LP-UV lamp. Using in silico calculations, we found that the *R*_AE_ of the 260 nm UV-LED (*R*_AE_ = 68.6) was higher than that of the LP-UV lamp (*R*_AE_ = 11.1), and the *R*_AE_ of UV irradiations was strongly correlated to the viral inactivation effect by UV irradiations (Table 2). To increase the *R*_AE_ score, we developed a hybrid UV-LED (*R*_AE_ = 86.3) by combining three different UV-LEDs (Figure 7a). Irradiation by the hybrid UV-LED had the highest inactivation effect on both H1N1 and H6N2 subtypes among the UV-LEDs and LP-UV lamp at 4.8 mJ/cm^2^ fluence in this study (Figure 5). Beck S.E. and her colleagues previously showed that the inactivation effects of a combination of 260 nm and 280 nm UV-LEDs on MS2 bacteriophages, human adenovirus 2, and four human enteric viruses were not significantly different than that of the 260 nm or 280 nm UV-LED alone [28]. Because the fluence rate of the 280 nm UV-LED was higher than that of the 260 nm UV-LED [28], the combined irradiation might not be enough to increase the *R*_AE_ score. These results suggest that the peak WLs of UV-LEDs and the ratio of fluence rate of UV-LEDs combined may be important factors to increase the *R*_AE_ score.

We investigated the correlations between the viral inactivation effect by UV irradiations and the *R*_AE_ scores of the irradiations in this study. However, the correlations between the bactericidal effects and the scores remain to be elucidated. We previously reported that a combination of 365 nm UV-LED and LP-UV lamp irradiation exhibited a synergistic bactericidal effect on *Vibrio parahaemolyticus* that was dependent on the suppression of cyclobutene pyrimidine dimer (CPD) repair, such as recA- and lexA-mediated SOS responses. [29]. Our previous data were supported by Xiao Y., et al. [30] and Song K., et al. [31] who showed some synergistic inactivation effects on *E. coli* by the suppression of DNA repair using combined 265/365 nm UV-LED irradiations. However, some reports showed that the combination of 265/280 nm UV-LEDs had no synergistic effect on *E. coli* [17,31]. Irradiation by 365 nm (*R*_AE_ = 0.253) showed a minimal contribution to the increased *R*_AE_ score (Table 1). These results suggest that the inactivation effects of combined UV-LEDs with different peak WLs may differ between bacteria and viruses.

We showed the loss of infectivity by UV-LED irradiations was due to damage to viral RNA, confirmed by RT-PCR, as previously reported in poliovirus-1 and bacteriophages [20,24,25]. Determinate damage of viral RNA was not, however, defined in both those reports and this study. Pyrimidines are more sensitive bases against UV irradiation than purines [32]. UV irradiation generates not only DNA photoproducts such as CPDs, 6-4 pyrimidine–pyrimidone (6-4PP), and cytosine/thymine hydrate [32], but also several RNA photoproducts such as uracil cyclobutane dimer and uracil/cytosine hydrate [33]. We had determined irradiations by 260, 270, and 280 nm UV-LED at the same fluence (4.8 mJ/cm^2^) in this study, and generated CPDs in herpes simplex virus 1 by dot-blot analysis using anti-CPDs antibody (data not shown). Petit-Frère C, et al. [34] reported that the photoproducts, including CPDs and 6-4PP, suppressed both DNA and RNA synthesis. From these reports, UV-LED-induced damage to viral RNA might be dependent on the formation of RNA photoproducts in IAVs. Unlike viral RNA, HA titer of H1N1 subtype was not changed by UV irradiations in this study (Figure 3). HA is a target for vaccines for IAVs because antibodies against HA are a major component of the human immune response and influenza vaccination [35]. Some studies showed that UV-inactivated viruses, including coronaviruses, were useful for the vaccines [36,37], but the studies about vaccines using UV-inactivated influenza viruses were scarcely reported [38]. Therefore, further work is needed to confirm whether UV-LED is useful for the generation of influenza inactivated vaccines.

We compared the viral inactivation effects of different UV-LEDs on various IAV subtypes, including H1N1 and avian H6N2, infected into MDCK cells and embryonated chicken eggs. In infected MDCK cells, both H1N1 and H6N2 subtypes were equally inactivated by UV-LED irradiations (Figure 7c,d). Similarly, Szeto W. et al. [39] demonstrated that H1N1 and H3N2 subtypes were equally inactivated by a vacuum-UV lamp for infection into MDCK cells. Meanwhile, in infected embryonated chicken eggs, the inactivation of H6N2 by UV-LED irradiation was reduced compared with the H1N1 subtype in this study. However, the crucial factors responsible for the different IAV strain-specific sensitivities to UV irradiation remain to be elucidated. Sutejo R., et al. [40] reported that the replication of the H1N1 subtype in chicken embryonic fibroblasts was lower than that of HPAI or LPAI. These data suggest that differences in the host responses to H1N1 and avian viruses may be a factor contributing to the subtype sensitivities to UV irradiations. Compared with IAVs and the other viruses, including MS2 coliphage, human adenovirus type 2 and human enteric viruses showed a lower sensitivity to the virucidal effect, by 260 nm or 280 nm UV-LED irradiation, respectively, than IAVs [19,28,41]. These data suggest that the sensitivities of various virus species and subtypes to UV-LED irradiation are different, and thus other strategies of UV-LED irradiation are necessary to inactivate UV-resistant subtypes.

In this study, we propose that the *R*_AE_ score should be an important factor for increasing the virucidal effect by UV-LED irradiation. However, we calculated the score using a range of 254–365 nm as the peak WL of the UV-LEDs and LP-UV lamp, but we should improve the *R*_AE_ score by generating more data using lower range WLs of the UV irradiations. Our study may contribute to preventing the spread of IAVs and avian flu viruses. Because the external quantum efficiencies of AlGaN-based UV-LEDs remain <10%, the development of higher quantum efficiencies is necessary for the application of viral inactivation in the near future.

## Figures and Tables

**Figure 1 microorganisms-08-01014-f001:**
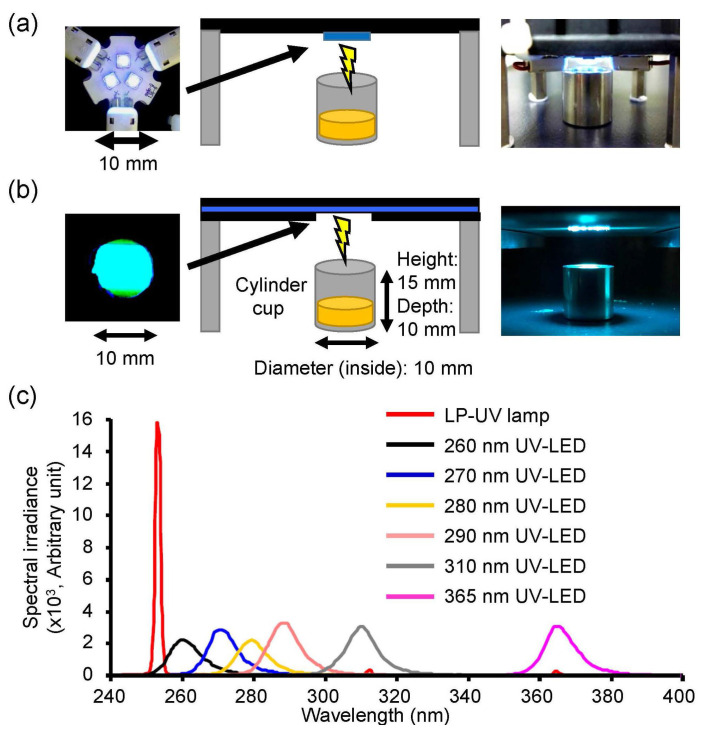
Irradiation system setup and emission spectrum of the UV-LEDs and low pressure-mercury (LP)-UV lamp used in this study. Photographs and schematic images of the UV-LEDs (**a**) and LP-UV lamp (**b**). (**c**) Emission spectrum of UV-LEDs. Detailed conditions of UV-LED irradiations are listed in Table 1.

**Figure 2 microorganisms-08-01014-f002:**
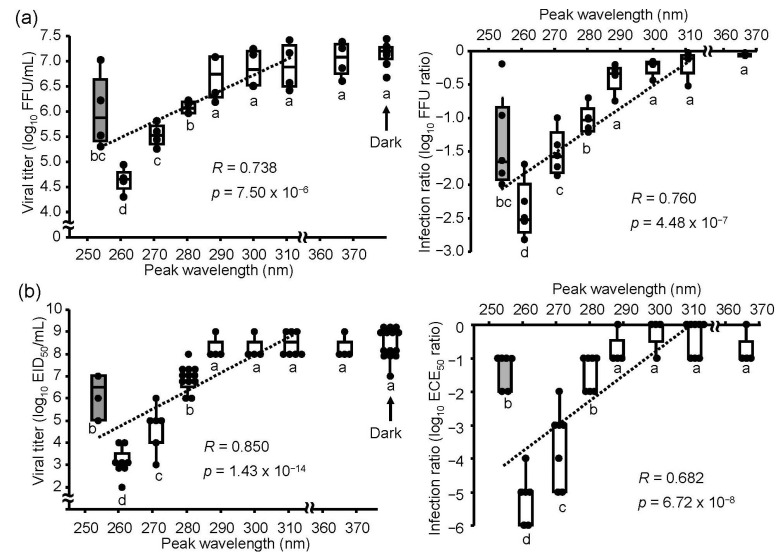
Inactivation effect of the UV-LEDs and LP-UV lamp on the influenza A virus H1N1 subtype. Viral suspensions of the H1N1 subtype (strain A/Puerto Rico/8/1934) were irradiated by different peak wavelength UV-LEDs (empty boxes) and an LP-UV lamp (gray box) at 4.8 mJ/cm^2^ and infected into MDCK cells (**a**) or embryonated chicken eggs (**b**). Viral inactivation effects of these irradiations were determined by the difference of viral titers (left panels) and the log_10_ focus-forming assay (FFU) ratio and embryo infection dose (EID_50_) ratio (right panels), as described in the Materials and Methods section. Each box plot shows the median (dark bar), values to the 1.5 interquartile ranges (whiskers), and 25 to 75 percentile ranges (box) (*n* = 4–8, *n* = number of independent replicates). The dashed line shows the relationship between peak wavelengths and infection ratios. Different letters of the alphabet indicate a statistical difference (*p* < 0.05) compared with each other.

**Figure 3 microorganisms-08-01014-f003:**
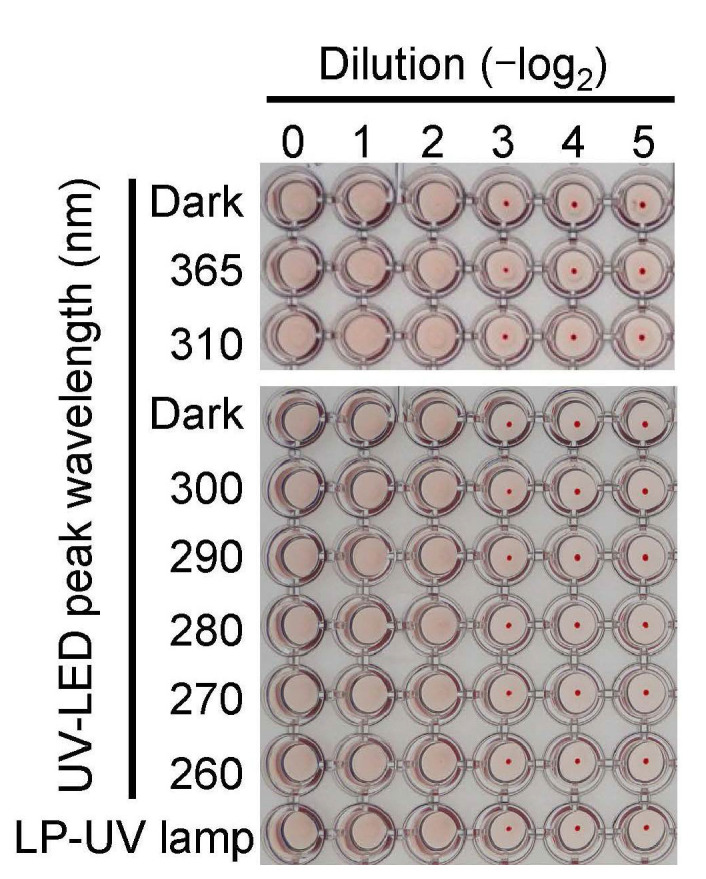
Effect of UV-LED irradiation on the hemagglutination titer. Viral suspensions of the H1N1 subtype (strain A/Puerto Rico/8/1934) were irradiated by UV-LEDs and the LP-UV lamp at 4.8 mJ/cm^2^, and the hemagglutination activity was determined as described in the Materials and Methods section.

**Figure 4 microorganisms-08-01014-f004:**
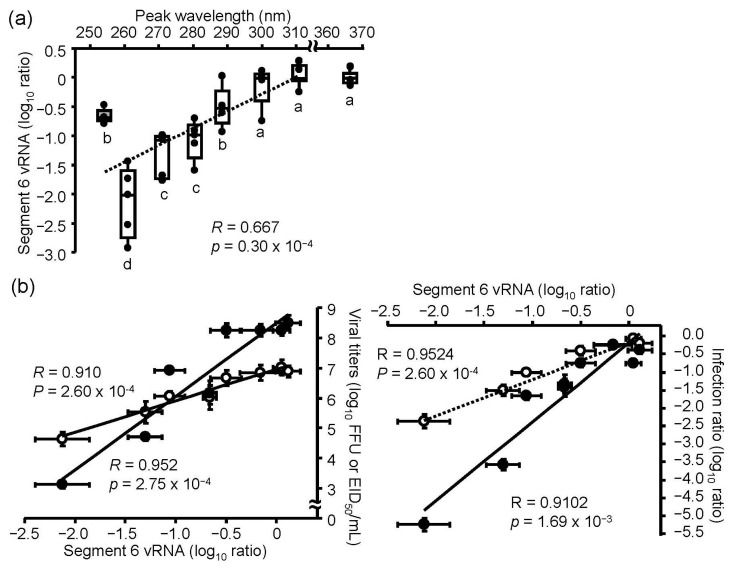
Effect of UV-LED irradiation on viral RNA. Viral suspensions of the H1N1 subtype (strain A/Puerto Rico/8/1934) were irradiated by UV-LEDs or an LP-UV lamp. Viral RNA damage was measured by vRNA (segment 6) strand-specific RT-qPCR, as described in the Materials and Methods section. (**a**) Damage of viral RNA by each UV irradiation. The dashed line shows the relationship between the peak wavelength and infection ratio. Different letters of the alphabet indicate a statistical difference (*p* < 0.05) between each other (*n* = 4–6, *n* = number of independent replicates). (**b**) The relationship between the inactivation effects and damage of viral RNA by each UV irradiation. Open circles and the dashed line indicate the data of the infection to MDCK cells. Filled circles and the solid line indicate the infection to embryonated chicken eggs. Viral inactivation effects of these irradiations were indicated by the difference of viral titers (left panels) and the log_10_ focus-forming assay (FFU) ratio and embryo infection dose (EID_50_) ratio (right panels), as described in the Materials and Methods section. Results are displayed as means ± SE.

**Figure 5 microorganisms-08-01014-f005:**
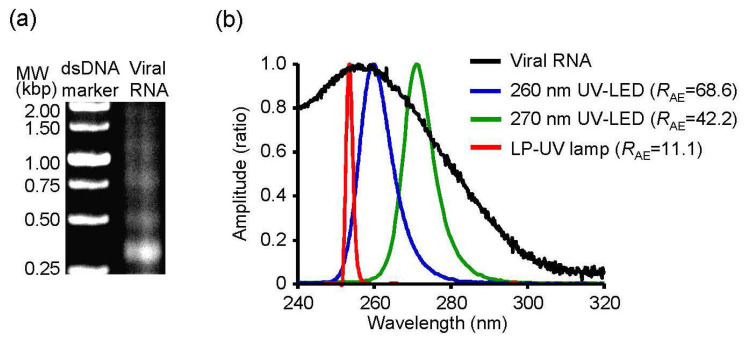
Absorbance spectrum of viral RNA and emission spectrum of UV irradiations. (**a**) A representative image of agarose gel electrophoresis for purified viral RNA. (**b**) Absorbance spectrum of viral RNA and emission spectrum of UV irradiations. *R*_AE_ score indicates the correlation coefficient between the absorbance spectrum of viral RNA and emission spectrums of UV irradiations.

**Figure 6 microorganisms-08-01014-f006:**
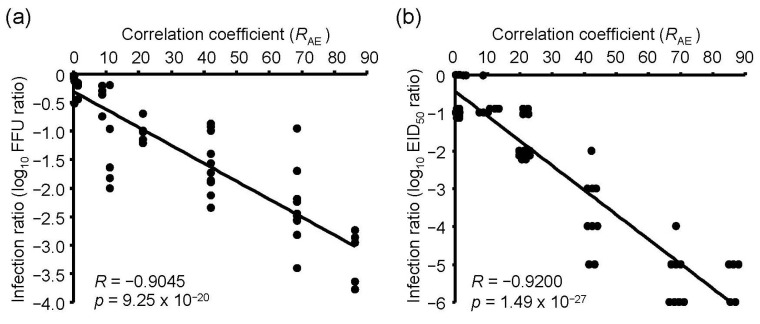
The relationship between the inactivation effects of UV irradiations and *R*_AE_ scores. Inactivation effects were measured by infection into MDCK cells (**a**) or embryonated chicken eggs (**b**) as host organisms. *R*_AE_ score indicates the correlation coefficient between the absorbance spectrum of viral RNA and emission spectrums of UV irradiations.

**Figure 7 microorganisms-08-01014-f007:**
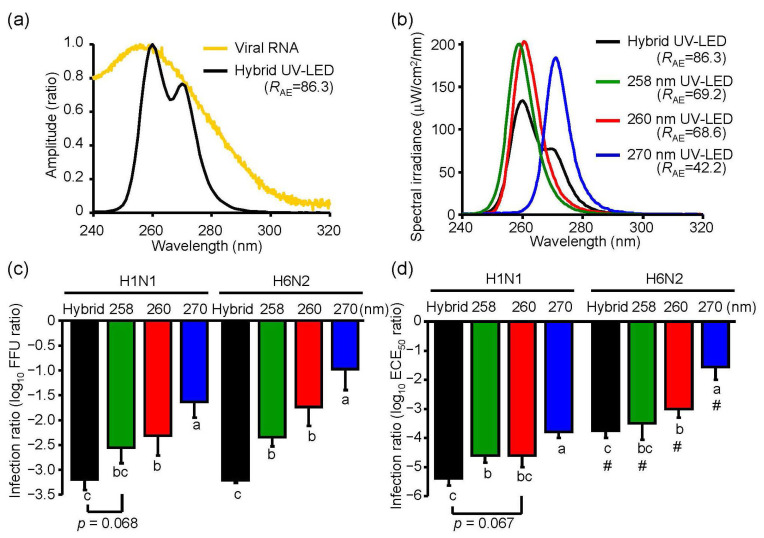
Inactivation effect of the hybrid UV-LED on influenza A virus H1N1 and H6N2 subtypes. (**a**) Absorbance spectrum of viral RNA and emission spectrum of hybrid UV-LED. (**b**) Emission spectrums of the hybrid UV-LED, 258 nm UV-LED, 260 nm UV-LED, and 270 nm UV-LED at 4.8 mJ/cm^2^. *R_AE_* score indicates the correlation coefficient between the absorbance spectrum of viral RNA and emission spectrums of UV irradiations. Detailed conditions of the hybrid UV-LED were described in Section 3.5. (**c**,**d**) Inactivation effect of the hybrid LED on influenza A virus H1N1 and H6N2 subtypes. Viral suspensions of H1N1 (strain A/Puerto Rico/8/1934) or H6N2 (A/Duck/Hong Kong/960/1980) were irradiated by the UV-LEDs at 4.8 mJ/cm^2^ and infected into MDCK cells. (**c**) or embryonated chicken eggs. (**d**). Results are displayed as means ± SE (*n* = 5–8, *n* = number of independent replicates). Different letters of the alphabet indicate a statistical difference (*p* < 0.05) between each other. (#) indicates *p* < 0.05 vs. H1N1 subtype under the identical UV-LED irradiation.

**Table 1 microorganisms-08-01014-t001:** Characteristics and Irradiating Conditions of the UV-light emitting diodes (UV-LEDs) and Low Pressure (LP)-UV Lamp Used in This Study.

Model Name	Peak WL, Spec (nm)	Peak WL, Measured (nm)	IF (A)	Fluence Rate (mW/cm^2^)	*R* _AE_
NVSU233A-U365	365	366.6	0.042	2.4	0.2525
NVSU234A-U310	310	310.9	0.136	2.4	0.3461
MO-2257-U300	300	300.3	0.150	2.4	1.4436
MO-2257-U290	290	288.5	0.133	2.4	8.8498
NVSU234A-U280	280	280.5	0.160	2.4	21.4084
MO-2257-U270	270	271.0	0.150	2.4	42.2429
MO-2257-U260	260	261.0	0.350	2.4	68.5566
UVC-S212T5 (LP-UV lamp)	254	254.0	-	2.4	11.1486

Note: WL, wavelength; IF, forward current; *R*_AE_, correlation coefficient between absorbance spectrum of viral RNA and emission spectrum of UV irradiations; LP-UV lamp, mercury low pressure ultraviolet lamp.

**Table 2 microorganisms-08-01014-t002:** The Difference in Correlation Coefficients between the Disinfection Ratio and Peak Wavelength or *R*_AE_ of UV Irradiations.

Host Organism. for Infection	*R*_N_ (Disinfection Ratio – Peak WL of UV Irradiations)	*R*_N_ (Disinfection Ratio – *R*_AE_ of UV Irradiations)	*p* Value
MDCK cells	0.760	0.9045	0.0488
embryonated chicken eggs	0.682	0.9200	0.000087

WL, wavelength; *R*_AE_, correlation coefficient between the absorbance spectrum of viral RNA and emission spectrum of UV irradiations; *R*_N_, natural number of correlation coefficients between the disinfection ratio and peak WL or *R*_AE_ of UV irradiations.

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
