# Peer review of "Irradiation by a Combination of Different Peak-Wavelength Ultraviolet-Light Emitting Diodes Enhances the Inactivation of Influenza A Viruses"

_microorganisms, 2020, doi:10.3390/microorganisms8071014_

Round 1

Reviewer 1 Report

Influenza A virus (IAV) is an important human respiratory pathogen that cause seasonal epidemics and occasional pandemics. Ultraviolet radiation (UV) disinfection technology is of growing interest since it was demonstrated that UV radiation is very effective against multiple viruses and pathogens, including IAV. The establishment of certain guidelines and protocols for IAV inactivation would be highly important in terms of control and prevention of transmission. Previously, UV has been demonstrated to be an effective approach to inactivate viruses (including IAV) through damage of their genome. In this manuscript “Irradiation by a combination of different peak wavelength ultraviolet-light emitting diodes enhances the inactivation of influenza A viruses” Authors have evaluated several irradiation protocols to optimize IAV inactivation. In addition, authors have established a hybrid UV-LED protocol as an efficient method for viral inactivation. Importantly, it has been suggested that Rae score is a factor that should be considered for increasing the anti-IAV effect of UV irradiation. Overall, this is a well written and interesting document. However, to my opinion, there are some major and minor concerns that the authors should consider to improve the quality of the document.

Major concerns:

  • Why different dose of H1N1 and H6N2 viruses have been used in the experiments?
  • Authors should include some references for Rae score? Moreover, If Rae factor (and the formula used) have been used in similar articles, this should be indicated.
  • Authors have used RT-qPCR to determine the damage to viral RNA by UV irradiation. Authors should include references where RT-qPCR have been used previously to determine RNA damage.
  • In figure 2 and 4, authors have represented the viral titers ratios. However, a more typical representation including the viral titers (Log10 FFU/ML) should be included (in addition to the previous ratio determination) for a better visualization of viral titers differences.
  • What means damage of the RNA? RNA cleavage, mutation…
  • In section 3.4, authors have indicated that viral RNA from the chorioallantoic fluid in IAV–infected eggs was purified. However, the protocol for this procedure have not been described.
  • Figure 5a: Authors should include a control of mock-infected eggs. It is unclear how authors determined that the observed bands correspond to viral RNA. Thus, this experiment should be done with the appropriate controls.
  • Figure 5b: include 270 nm.
  • Hybrid UV-led shown different levels of inactivation for H1N1 and H6N2 viruses. Have the authors an explanation for that? Since the composition of both viruses is similar, this result is surprising.
  • Authors can consider include supplementary material in the main text.

Minor concerns:

Line 46. Currently there are four types of influenza viruses (IAV, IBV, ICV and IDV).

Line 206. IVA should be IAV. Please check the document for other typos.

Authors could consider comment if the described technology could be used to generate influenza inactivated vaccines.

Author Response

Thank you for your valuable comments. We answered the reviewer’s comments as described below.

Q1. Why different dose of H1N1 and H6N2 viruses have been used in the experiments?

A1. We understood we should irradiate equal virus titers of both subtypes. We prepared and irradiated approximately twice higher viral titer of H1N1 subtype (1.92 ± 0.13 × 10^7 FFU/mL) than that of H1N1 (0.83 ± 0.05 × 10^7 FFU/mL). Because H6N2 subtype was propagated lower than H1N1 in 10-day-old chicken embryonated eggs for 48 h at 37°C, we could not increase the viral titer of H6N2 up to H1N1. In infected to MDCK cells, the infection ratios with irradiation by the hybrid or the other independent UV-LED were not significant difference between H1N1 and H6N2 (Figure 7c), suggesting that the difference of viral titers between the subtypes in this study might be not important factor for the effects of UV irradiations.

Q2. Authors should include some references for Rae score? Moreover, If Rae factor (and the formula used) have been used in similar articles, this should be indicated.

A2. We could not find any articles about the RAE or the related scores/factors because we originally developed the RAE factor and the formula in this study. We referred some articles those reported virucidal effects by combination of different peak-wavelength UV-LEDs in the Discussion section, but those showed the combinations did not increase virucidal effects by comparison with without the combination. For example, Beck SE and her colleagues previously reported that the inactivation effects of a combination of 260 nm and 280 nm UV-LEDs on MS2 bacteriophage, human adenovirus 2, and four human enteric viruses were not significantly different than that of the 260 nm or 280 nm UV-LED alone [reference #32]. We simulated the estimated RAE score of the combined irradiation in the report (Figure A), but the score (RAE = 58.4) was lower than that of 260 nm alone. Because the fluence rate of the 280 nm UV-LED was higher than that of the 260 nm UV-LED, the combined irradiation might be not enough to increase the RAE score. These results suggest that both the peak WLs and the ratio of fluence rate of UV-LEDs may be important factors to increase the RAE score by combination of UV-LEDs. We inserted the sentence about the the RAE score of previous reports in the end of first paragraph in the Discussion section.

Q3. Authors have used RT-qPCR to determine the damage to viral RNA by UV irradiation. Authors should include references where RT-qPCR have been used previously to determine RNA damage.

A3. Simonet J, et al. and Beck SE, et al. had been previously used to determine viral RNA damage in poliovirus 1 and bacteriophages [reference #28 and #29]. We insert the references about RT-qPCR for viral RNA damage in the Section 2.6.

Q4. In figure 2 and 4, authors have represented the viral titers ratios. However, a more typical representation including the viral titers (Log10 FFU/ML) should be included (in addition to the previous ratio determination) for a better visualization of viral titers differences.

A4. We agreed your comment about visualizations of viral titer differences in Figures 2 and 4. We added the figures including the viral titers (Log10 FFU/mL or Log10 EID50/mL) in Figures 2 and 4b of the revised manuscript.

Q5. What means damage of the RNA? RNA cleavage, mutation…

A5. Previous researches showed the loss of infectivity by UV irradiations was due to damage to viral RNA confirmed by RT-PCR and RNA damage in poliovirus-1, bacteriophages, and IAV [reference #24, 28, 29]. However determinate damage of viral RNA was not defined in both those reports and this study. Pyrimidines are more sensitive bases against UV irradiation than purines [reference #36]. UV irradiation generates not only DNA photoproducts as cyclobutane pyrimidine dimmers (CPDs), 6–4 pyrimidine–pyrimidone (6–4PP), and cytosine/thymine hydrate [reference #36] but also several RNA photoproducts such as uracil cyclobutane dimer and uracil/cytosine hydrate [reference #37]. We had determined irradiations by 260, 270, and 280 nm UV-LED at the same fluence (4.8 mJ/cm2) in this study generated CPDs in herpes simplex virus 1 by dot-blot analysis using anti-CPDs antibody (Figure B, unsubmitted data). Petit-Frère C, et al. reported that the photoproducts including CPDs and 6-4PP suppressed both DNA and RNA synthesis [reference #38]. From these reports, UV-LED-induced damage to viral RNA might be depended on formation of RNA photoproducts in IAVs. We inserted the sentence about the damage of RNA by RT-PCR in third paragraph in the Discussion section.

Q6. In section 3.4, authors have indicated that viral RNA from the chorioallantoic fluid in IAV–infected eggs was purified. However, the protocol for this procedure have not been described.        

A6. We agreed your suggestion about the protocol for purification of viral RNA. We insert the sentence about the protocol in the Section 2.7, ‘‘After propagating the virus in 10-day-old chicken embryonated eggs for 48 h at 37°C, allantoic fluids were precleared by centrifugation at 3,300 g for 20 min followed by filtration through 0.45 mmf pore filters. The viruses were then purified by centrifugation (112,500 x g for 2 h) through PBS containing 20% sucrose, as previously reported [25]. Virus pellets were resuspended in PBS and and purified the viral RNA by a QIAamp Viral RNA Mini Kit (Qiagen). The purified viral RNA were checked by RT-PCR for segment 4 (supplementary Figure S1), as previously reported [24]’’

Q7. Figure 5a: Authors should include a control of mock-infected eggs. It is unclear how authors determined that the observed bands correspond to viral RNA. Thus, this experiment should be done with the appropriate controls.

A7. We agreed your comments about the appropriate controls for purification of viral RNA. We prepared mock or H1N1-infected chicken embryonated eggs and purified viral RNA. However we did not find any pellets after the ultracentrifusion and enough RNA concentration for agarose gel electrophoresis (supplementary Figure S1a in the revised manuscript). And we confirmed the observed bands correspond to viral RNA by RT-PCR for segment 4 (supplementary Figure S1b).

Q8. Figure 5b: include 270 nm.

A8. We added the emission spectrum of 270 nm UV-LED in Figure 5b of the revised manuscript.

Q9. Hybrid UV-led shown different levels of inactivation for H1N1 and H6N2 viruses. Have the authors an explanation for that? Since the composition of both viruses is similar, this result is surprising.

A9. As you pointed out, the inactivation of H6N2 by hybrid UV-LED was lower than that of H1N1 in infected embryonated chicken eggs (Figure 7d). Meanwhile, in infected MDCK cells, both H1N1 and H6N2 subtypes were equally inactivated by UV-LED irradiations (Figure 7c). However, the crucial factors responsible for the different IAV strain-specific sensitivities to UV irradiation remain to be elucidated. Sutejo R, et al. reported that the replication of the H1N1 subtype in chicken embryonic fibroblasts was lower than that of HPAI or LPAI [reference #44]. From the report, we thought that differences in the host responses to H1N1 and avian viruses may be a factor contributing to the subtype sensitivities to UV irradiations. The comments were inserted in the fourth paragraph of the Discussion section.

Q10. Authors can consider include supplementary material in the main text.

A10. We moved original ‘supplementary Figure S1’ into the body (Figure 6) of the revised manuscript in accordance with your suggestion.

Minor concerns:

Q11. Line 46. Currently there are four types of influenza viruses (IAV, IBV, ICV and IDV).

A11. We corrected the point in the revised manuscript.

Q12. Line 206. IVA should be IAV. Please check the document for other typos.

A12. We corrected ‘IVA’ into IAV in the revised manuscript.

Q13. Authors could consider comment if the described technology could be used to generate influenza inactivated vaccines.

A13. We inserted the comments about generation of influenza inactivated vaccines by UV-LED irradiations in third paragraph in the Discussion section, ‘‘Unlike viral RNA, HA titer of H1N1 subtype was not changed by UV irradiations in this study (Figure 3). HA is a target for vaccines for IAVs because antibodies against HA are a major component of the human immune response and influenza vaccination [39]. Some researches showed that UV-inactivated viruses including coronaviruses were useful for the vaccines [40,41], but the researches about vaccines using UV-inactivated influenza viruses were scarcely reported [42]. Therefore, further work is needed to confirm whether UV-LED is useful for generation of influenza inactivated vaccines.’’

Reviewer 2 Report

The manuscript is well written and clearly presented. The authors tested different UV-LEDs WL on H1NI and H6N2 strains of IAV and found that UV-LED at 260nm had the highest inactivation. Also found that a combination of UV-LEDs WL at 258, 260 and 270nm was more effective inactivating this virus that a single WL. 

However the combination of UV-LEDs includes a 258 nm WL that has not been tested alone, therefore the results of the combination of different WL is theoretical because we do not know if the UV-LED alone at 258 nm is as effective as the combinatio of the different WL together. My suggestion for the authors is to consider to perform the study working with the UV-LED 258 nm alone in order to have the complete validation. Otherwise, the authors should aware the readers that the data of the combination effect of different WL may reflect the effect of the 258nm alone. In nay case, the authors should include a statement in the article for this potential vias in their conclusion.

Author Response

Thank you for your valuable comments. We understood we should include the data of irradiation by 258 nm UV-LED alone. We added the data of irradiation by 258 nm UV-LED alone in Figure 6 of the revised manuscript. In infected MDCK cells, the hybrid LED had a significantly (P < 0.05) or a moderate trend toward significantly (P = 0.068) higher inactivation effect on both H1N1 and H6N2 subtypes than the 258 nm UV-LED (Figure 6c). In infected embryonated chicken eggs, the hybrid LED showed a significantly (P < 0.05) higher inactivation effect on H1N1 subtype than the 258 nm UV-LED, but the effect on H6N2 was not different between the hybrid LED and 258 nm LED (Figure 6d). As these results, we modified the Section 3.5 of the revised manuscript. And we moved the numbers of the reference after ‘et al.’ in accordance with your instructions. And we moved original ‘supplementary Figure S1’ into the body (Figure 6) of the revised manuscript. 

Round 2

Reviewer 1 Report

Authors have addressed all the comments from reviewers and they have improved the manuscript. I recommend the publication of the article in the present form.

Reviewer 2 Report

I appreciated that the authors include my suggestions in the revised version of the manuscript.